# Immunomodulatory and Antioxidant Properties of Wheat Gluten Protein Hydrolysates in Human Peripheral Blood Mononuclear Cells

**DOI:** 10.3390/nu12061673

**Published:** 2020-06-04

**Authors:** Ivan Cruz-Chamorro, Nuria Álvarez-Sánchez, Guillermo Santos-Sánchez, Justo Pedroche, María-Soledad Fernández-Pachón, Francisco Millán, María Carmen Millán-Linares, Patricia Judith Lardone, Ignacio Bejarano, Juan Miguel Guerrero, Antonio Carrillo-Vico

**Affiliations:** 1Instituto de Biomedicina de Sevilla, IBiS (Universidad de Sevilla, HUVR, Junta de Andalucía, CSIC), 41013 Seville, Spain; icruz-ibis@us.es (I.C.-C.); nalvarez-ibis@us.es (N.Á.-S.); gsantos-ibis@us.es (G.S.-S.); plardone@us.es (P.J.L.); ibejarano@us.es (I.B.); guerrero@us.es (J.M.G.); 2Departamento de Bioquímica Médica y Biología Molecular e Inmunología, Universidad de Sevilla, 41009 Seville, Spain; 3Plant Protein Group, Instituto de la Grasa, CSIC, 41013 Seville, Spain; j.pedroche@csic.es (J.P.); fmillanr@ig.csic.es (F.M.); 4Área de Nutrición y Bromatología, Departamento de Biología Molecular e Ingeniería Bioquímica, Universidad Pablo de Olavide, Ctra. Utrera Km 1, 41013 Sevilla, Spain; msferpac@upo.es; 5Cell Biology Unit, Instituto de la Grasa, CSIC, 41013 Seville, Spain; mcmillan@ig.csic.es; 6Departamento de Bioquímica Clínica, Unidad de Gestión de Laboratorios, Hospital Universitario Virgen del Rocío, 41013 Seville, Spain

**Keywords:** wheat gluten protein hydrolysates, bioactive peptides, pro-inflammatory cytokines, oxidative stress, antioxidant capacity, glutathione, peripheral blood mononuclear cells

## Abstract

Peptides from several plant food proteins not only maintain the nutritional values of the original protein and decrease the environmental impact of animal agriculture, but also exert biological activities with significant health-beneficial effects. Wheat is the most important food grain source in the world. However, negative attention on wheat-based products has arose due to the role of gluten in celiac disease. A controlled enzymatic hydrolysis could reduce the antigenicity of wheat gluten protein hydrolysates (WGPHs). Therefore, the aims of the present study were to evaluate the effects of the in vitro administration of Alcalase-generated WGPHs on the immunological and antioxidant responses of human peripheral blood mononuclear cells (PBMCs) from 39 healthy subjects. WGPH treatment reduced cell proliferation and the production of the Type 1 T helper (Th1) and Th17 pro-inflammatory cytokines IFN-γ and IL-17, respectively. WPGHs also improved the cellular anti-inflammatory microenvironment, increasing Th2/Th1 and Th2/Th17 balances. Additionally, WGPHs improved global antioxidant capacity, increased levels of the reduced form of glutathione and reduced nitric oxide production. These findings, not previously reported, highlight the beneficial capacity of these vegetable protein hydrolysates, which might represent an effective alternative in functional food generation.

## 1. Introduction

Imbalanced diets low in fruits, vegetables, nuts, and whole grains and high in red and processed meat are the principal risk factors implicated in the development of some diseases [1]. Thus, diet is an important adaptable risk factor in the prevention of obesity, diabetes, hypertension, and other cardiovascular diseases components [2]. Diets which include plant foods, with special emphasis on plant protein sources, provide a healthy dietary pattern to reduce cardiovascular diseases [3,4]. The consumption of functional food from natural vegetable sources is an interesting strategy not only for replacing animal proteins with plant proteins, which is a central issue for health [5], but also for reducing the environmental impact of animal agriculture [6]. Wheat, along with rice and maize, is the most important cereal consumed by humans [7]. The main components in wheat flour are gluten, starch and non-gluten α-amylase/trypsin inhibitor (ATI) family. Unlike starch, gluten can trigger celiac disease (CD), an immune-mediated gastrointestinal disorder [8], while ATI can trigger the innate immune response [9]. A controlled enzymatic hydrolysis could reduce the antigenicity of wheat gluten protein hydrolysates (WGPHs) [10,11,12]. Moreover, the hydrolysis process not only generates peptides that maintain the nutritional values of the original protein, but can also give rise to bioactive peptides with remarkable beneficial health effects [13,14]. In this way, a large body of evidence supports biological activities mediated by animal or vegetable hydrolysates, including immunomodulatory, antioxidant, anticancer, or antibacterial functions [15,16]. However, most of these data come from experimental approaches based on cell-free systems or immortalized cell lines [15].

Immune response and the control of oxidative stress are two interconnected processes implicated in the maintenance of homeostasis [17]. Although inflammation is a physiological reaction, the exacerbated cytokine-mediated response can trigger several pathological conditions. Thus, an increase of pro-inflammatory Type 1 T helper (Th1) and/or Th17 cytokines, such as interferon-γ (IFN-γ) and interleukin (IL)-17, respectively, is involved in the generation of several inflammatory diseases, such as the inflammatory bowel disease and CD [18,19]. On the contrary, Th2 anti-inflammatory cytokines, including IL-10 and IL-4, are implicated in the control of pro-inflammatory responses [18,20]. Hence, the fine control of the cellular pro-/anti-inflammatory microenvironment is crucial in health maintenance. In addition to the immune response, the antioxidant system also participates in maintaining the basal status of the organism. Although there is controversial information about the efficacy of antioxidant supplementation in ongoing severe diseases [21,22], an impairment of the oxidative balance has been described to play a pivotal role in the etiopathogenesis of different disorders, including CD [23,24]. The antioxidant response is mediated by enzymatic and non-enzymatic systems. The principal antioxidant enzymes implicated in the conversion of free-radicals into non-toxic compounds are superoxide dismutase (SOD), catalase (CAT) and glutathione peroxidase (GPx) [25], whereas the main non-enzymatic antioxidant is the reduced glutathione (GSH) that neutralizes free-radicals by direct transfer of an electron [26]. After reducing pro-oxidants compounds, two GSH molecules give rise to a molecule of glutathione disulfide (GSSG) that is potentially toxic to the cells. The glutathione reductase (GR) enzyme is responsible for regenerating GSH from GSSG. Therefore, the GSH/GSSG ratio is considered a powerful index of oxidative stress and disease risk [27].

Despite previous studies having described the effects of wheat peptides, derived from simulated digestion, on the immune response of monocytes/macrophages from healthy and CD subjects [28,29], to our knowledge, no study regarding the bioactive effects of WGPHs, a taylor-made protein hydrolysate obtained with the commercial protease Alcalase, on the whole fraction of human circulating blood leukocytes, has been reported. Thus, in the present study, we used an ex vivo approach of cultured human peripheral blood mononuclear cells (PBMCs) from healthy subjects to explore the immunomodulatory and antioxidant properties of WGPHs.

## 2. Materials and Methods

### 2.1. Characterization and Preparation of Wheat Gluten Protein Hydrolysates

The protein hydrolysates used in this study were obtained at Instituto de la Grasa (Consejo Superior de Investigaciones Científicas, Seville, Spain). Wheat gluten, supplied by Roquette (Vital wheat gluten, Roquette, Norte-Paso de Calais, France), was hydrolysed in distilled water (10% *w*/*v*) by means of the proteolytic enzyme Alcalase 2.4 L (E/S = 0.1 AU/g protein) for 60 min, at pH 8 and 50 °C. The enzyme was then inactivated by lowering to pH 4. WGPHs were recovered by centrifugation at 8000 rpm for 15 min and the pellet was discarded. WGPHs were frozen overnight and then lyophilized and stored at 4 °C. The chemical composition and degree of hydrolysis are shown in Table 1 and Figure 1a, respectively. The molecular profile of WGPHs is shown in Figure 1b. Protein concentrations, moisture, ash content, total dietary fiber, digestibility, and molecular profile were determined as previously described [30]. WGPHs were dissolved in incomplete RPMI 1640 medium (BioWest, Nuaillé, France) before each experiment. When completely dissolved, they were filtered through a sterile membrane filter (0.2 µm pore size) and then autoclaved.

### 2.2. Cell Culture

PBMCs were obtained from fasting blood samples provided anonymously from 39 healthy adult volunteers who signed an informed consent form. The study followed the Helsinki Declaration for medical research involving human subjects and was approved by the Virgen Macarena-Virgen del Rocío University Hospital ethical committee (reference number 2012PI/200). Blood samples were collected in BD Vacutainer^®^ CPT™ Mononuclear Cell Preparation Tubes (BD Biosciences, San Jose, CA, USA) containing sodium heparin, and PBMCs were isolated immediately by centrifugation. Cells were then cultured at 1 × 10^6^ cells/mL in RPMI 1640 medium supplemented with 10% fetal bovine serum, 2 mM L-glutamine, 100 U/mL penicillin and 100 U/mL streptomycin (all from Biowest) in the presence or absence of two different concentrations of WGPHs (0.25 and 0.5 mg/mL) and incubated at 37 °C in a 5% CO_2_ humidified atmosphere. In addition, PBMCs were stimulated with 8 µg/mL of phytohaemagglutinin-P (PHA) (Sigma-Aldrich, St. Louis, MO, USA), a T cell proliferative stimulus, a T cell proliferative stimulus, with or without WGPHs to analyze proliferation and cytokine production.

### 2.3. Cell Proliferation and Viability Assays

Cell proliferation induced by the mitogenic agent PHA was determined after 72 h of cell culture (100,000 cells/well) by the 5-bromo-2-deoxyuridine (BrdU) Cell Proliferation ELISA (Roche Diagnostic, Basel, Switzerland), according to the manufacturer’s instructions. This assay is based on the measurement of modified deoxyuridine incorporation during DNA synthesis. Cell viability was determined by adding the Cell Proliferation Reagent WST-1 (Roche) to the cultures for the last 5 h of culture. This assay is based on the cleavage of WST-1 to a fluorescent compound mediated by metabolically active (live) cells. For both assays, absorbance was measured at 450 nm (reference wavelength: 620 nm) with a Multiskan™ FC Microplate Photometer (Thermo Scientific, Vantaa, Finland). A blank control was used to subtract background absorbance from all the samples.

### 2.4. Cytokine Determination

Cytokine production was determined in the supernatants of 48-h PHA-stimulated cell cultures by the Human Kit FlowCytomix (eBioscience, San Diego, CA, USA) according to the manufacturer’s instructions. The analysis was carried out in a BD FACSCanto II Flow Cytometer (BD Biosciences).

### 2.5. RNA Extraction, Reverse Transcription and Real-Time PCR

Tripure Isolation Reagent (Roche) was used to extract RNA from stimulated or non-stimulated PBMCs incubated overnight with/without WGPHs, according to the manufacturer’s instructions. After isolation, RNA was converted to single-strand cDNA using Transcriptor First Strand cDNA Synthesis Kit (Roche). Real-time PCR was performed on a LightCycler^®^ 480 (Roche) using LightCycler^®^ 480 SYBR Green I Master (Roche). Primer sequences are detailed in Appendix A. The expression levels of each mRNA were normalized to that of β-actin, and the relative expression levels were calculated using the 2^−ΔΔCt^ method.

### 2.6. Enzymatic Activity Assays

The enzymatic activities of SOD (Arbor Assays, Ann Arbor, MI, USA), CAT, GPx (Cayman Chemical, Ann Arbor, MI, USA), GR (BioVision, Milpitas, CA, USA) and the Total Antioxidant Capacity (TAC) (Cell Biolabs, San Diego, CA, USA) were determined in supernatants from non-stimulated cells cultured overnight in the presence/absence of WGPHs according to manufacturer’s instructions. Antioxidant status was also studied by oxygen radical absorbance capacity (ORAC), ferric reducing antioxidant power (FRAP) and trolox equivalent antioxidant capacity (TEAC) assays, as previously described [31]. Samples were diluted 1:400 in phosphate buffer (75 mM, pH 7.4) and 1:15 in distilled water for ORAC and TEAC assays, respectively. GSH levels were quantified by Glutathione Assay Kit (Cayman Chemical), according to the manufacturer’s instructions. Moreover, nitric oxide (NO) levels were quantified by the Griess test (Sigma-Aldrich). Absorbances for NO, SOD, CAT, GPx, GR, and TAC were read with the CLARIOstar^®^ microplate reader (BMG labtech, Ortenberg, Germany). ORAC, FRAP, TEAC, and GSH absorbances were quantified by the Synergy™ HT-multimode microplate reader (Biotek Instruments, Winooski, VT, USA).

### 2.7. Statistical Analysis

Results were expressed as the mean and standard error of the mean (SEM) from at least five independent experiments. Data were analyzed with IBM^®^ SPSS^®^ Statistic software v24 (IBM, Armonk, NY, USA), using the paired Wilcoxon test. Given that data were not normally distributed, Bonferroni’s correction were applied. *p*-values ≤ 0.05 were considered statically significant.

## 3. Results

### 3.1. WGPHs Reduce Cell Proliferation without Modifying Cell Viability in PBMCs

WGPHs produced a significant dose-dependent decline in PHA-induced cell proliferation compared with the control group (Figure 2a). To investigate whether this effect was related to WGPHs cytotoxic actions, a viability assay based on WST-1 was carried out. As shown in Figure 2b, no WGPHs effects on cell viability were observed with respect to the untreated group. As 0.5 mg/mL WGPHs showed a greater anti-proliferative effect without impairing cell viability, this WGPHs concentration was used in the following approaches.

### 3.2. WGPHs Do not Trigger an Inflammatory Response

To examine whether WGPHs would be able to act on the PBMCs inflammatory response, the production of Th1 (IFN-γ) and Th17 (IL-17) pro-inflammatory, and Th2 (IL-4, IL-10) anti-inflammatory cytokines was analyzed in cell culture supernatants from 48 h PHA-stimulated PBMCs. As shown in Figure 3, WGPHs decreased IFN-γ (Figure 3a), IL-17 (Figure 3b), and IL-10 production (Figure 3c), while IL-4 levels were unchanged (Figure 3d). Although IL-4 levels were not affected by WGPH treatment and IL-10 production was even reduced, a significant increase in the ratios of IL-4/IFN-γ (Figure 3e), IL-4/IL-17 (Figure 3f) and IL-10/IFN-γ (Figure 3g) was observed upon WGPHs treatment. Although WGPHs decreased the levels of IL-10 in the cell culture supernatant, interestingly, a significant increase in IL-10 mRNA was observed in PHA-stimulated PBMCs treated with WGPHs (Appendix A).

### 3.3. WGPHs Increase the PBMCs Antioxidant Capacity

To assess possible effects of WGPHs on the antioxidant system, we analyzed the mRNA expression levels and enzymatic activities of SOD, CAT, GPx and GR enzymes involved in the early phase of detoxification of radical oxygen species (ROS). Although an increasing trend was observed in mRNA levels (Figure 4a) and enzymatic activities (Figure 4b) of SOD, CAT, and GR in the presence of WGPHs, only GR mRNA levels were significantly higher after WGPH treatment. In accordance with this increase, WGPHs significantly increased the GSH levels (Figure 4c). Additionally, the global WGPHs effect on the antioxidant status was assessed by TAC, ORAC, FRAP, and TEAC assays in the supernatant of PBMCs treated with WGPHs. All parameters were significantly raised in the presence of WGPHs (Figure 4d).

### 3.4. WGPHs Reduce NO Production

To analyze whether the WGPHs affected the detoxification of reactive nitrogen species, inducible nitric oxide synthase (iNOS) gene expression and NO production were studied. WGPHs significantly reduced both the iNOS mRNA levels in PBMCs (Figure 5a) and the NO production in the supernatant of PHA-stimulated PBMCs (Figure 5b).

## 4. Discussion

This study describes the beneficial immunomodulatory and antioxidant properties of WGPHs, obtained by enzymatic hydrolysis with Alcalase 2.4 L, a food-grade non-specific endopeptidase, on human PBMCs from healthy donors. Previous studies have shown better solubility and emulsification of Alcalase 2.4 L-hydrolyzed wheat protein compared with other proteases such as pepsin or pancreatin [32]. We also reported the activity of this protease in the generation of bioactive peptides [30,33,34,35]. WGPHs treatment was able to reduce cell proliferation as well as improve the cellular anti-inflammatory microenvironment, increasing Th2/Th1 and Th2/Th17 balances, without being cytotoxic. Additionally, WGPHs improved the global antioxidant capacity, increased GSH levels and reduced the NO production. To our knowledge, these effects have not been previously reported.

WGPHs not only reduced PBMCs proliferation but also decreased the production of IFN-γ and IL-17 cytokines, two key mediators involved in many inflammatory conditions, such as obesity, diabetes, inflammatory bowel disease and CD [18,19]. Remarkably, although WGPHs had no effect on IL-4 production, a typical Th2 cytokine implicated in counteracting Th1 pathological response [18], and even decreased IL-10 production in PHA-stimulated PBMCs, WGPHs significantly increased the molar ratios of Th2/Th1 and Th2/Th17 cytokines skewing the anti-/pro-inflammatory responses toward an anti-inflammatory microenvironment. Although the detected reduction in IL-10 production could be related to a WGPHs-mediated decrease in Th2 response, since this cell population can produce IL-10 [20], WGPHs administration significantly increased the levels of IL-10 mRNA. Therefore, the decreased IL-10 production in the supernatant of PHA-stimulated PBMCs treated with WGPHs could be more related to its anti-proliferative activity than to a direct effect on mRNA gene expression. This is also supported by the increased ratio of IL-10/IFN-γ in the WGPHs-treated PBMCs. There are many examples of the immune properties of the vegetable compounds, such as soybean [36], maize [37], common bean [38], *Salvia plebeia* [39], or lupine [30]. However, to date, there is no data about the immunomodulatory effects of WGPHs on human T lymphocytes.

Growing evidence regarding the relationship between the immune response and oxidative stress has been shown. In this way, the physiological immune processes generate reactive molecules such as superoxide anion (O_2_^−^), a highly toxic free radical oxygen species (ROS) involved in many pathophysiological conditions [40]. SOD enzymes protect aerobic organisms through the dismutation of O_2_^−^ to molecular oxygen and hydrogen peroxide (H_2_O_2_), which is the precursor of the hydroxyl radical (OH), the most devastating ROS. H_2_O_2_ can be neutralized to water through two independent pathways mediated by the CAT and GPx enzymes. Whereas CAT directly converts H_2_O_2_ into the water, GPx requires the oxidation of the cofactor GSH to GSSG, which is subsequently reduced to GSH through the GR enzyme to reestablish the GSH/GSSG balance. Therefore, the GSH/GSSG ratio is the major cellular redox buffer and the maintenance of GSH levels is an essential biological process [41]. Previous studies have shown the antioxidant properties of Alcalase 2.4 L-digested wheat gluten hydrolysates in cell-free in vitro systems [42,43,44]. Furthermore, in vitro treatment of the PC12 rat adrenal medulla cell line with WGPHs (0.25–1 mg/mL) protects against H_2_O_2_-induced oxidative stress [45]. Additionally, the daily administration of wheat peptides for 30 days in a rat model of small intestinal damage induced by non-steroidal anti-inflammatory drugs cause beneficial effects on the mucous membrane of the small intestine by decreasing oxidative stress [46]. These facts are in accordance with our results that show the anti-oxidant capacity of WGPHs. We also observed a significant increase in the gene expression of the GR enzyme, as well as a trend towards the increase in mRNA levels and antioxidant activities of SOD and CAT after WGPHs administration. Moreover, we showed an increase in the GSH levels in WGPHs-treated PBMCs, which agrees with the high mRNA levels of GR, the enzyme implicated in the catalysis of GSSG to GSH. The antioxidant role of WGPHs is also supported by the increase of the total antioxidant capacity quantified by several approaches such as TAC, FRAP, ORAC, and TEAC, used to assess the free radical scavenging activity of biological samples. TAC [47] and FRAP [48] are assays based on the determination of the ability of a sample to reduce copper and iron, respectively. The ORAC assay quantifies the neutralizing capacity against peroxyl radical-induced oxidation [49], whereas TEAC evaluates the ability of the biological sample to protect a radical molecule from its oxidation [50]. Therefore, the high levels of TAC, FRAP, ORAC and TEAC in WGPHs-treated PBMCs indicate the considerable antioxidant power of these hydrolysates. Some studies have shown the antioxidant capacity of vegetable peptides, such as oats [51], rice [52], corn [53], and wheat [54] by using similar approaches. However, no previous work has described the capacity of WGPHs to neutralize the oxidation of both metal and organic molecules in human cells.

The cross-talk between inflammation and oxidative stress is also shown through a complex bidirectional communication circuit that shares a common biochemical language including molecules such as NO. NO overproduction can induce cell damage by means of a reactive nitrogen species generation [55], leading to several disorders [56]. Among the three isoforms of the nitric oxide synthases, the enzymes implicated in NO synthesis [55], only the iNOS is activated during the inflammatory response [55]. In fact, the iNOS promoter is controlled by the nuclear factor-κB signaling pathway induced by inflammation [57]. Besides counteracting the overproduction of PHA-induced pro-inflammatory cytokines and increasing the cellular antioxidant status, WGPHs not only decreased the iNOS mRNA levels in PHA-stimulated PBMCs but also were able to reduce the NO overproduction. In agreement with our results, pyroglutamyl leucine, a peptide isolated from wheat gluten hydrolysate, has been reported to inhibit the induction of iNOS expression in primary rat hepatocytes cultures [58]. Although some other studies have shown that vegetable protein hydrolysates, such as soybean [59], amaranth [60], and lupine [33], can reduce NO production in RAW 264.7 or THP-1 macrophages, to our knowledge, the effect of WGPHs on NO production has not been previously described in human cells.

Even though the lack of characterization of peptides from the WGPHs could be considered a limitation of the study, the main aim of the present work resided in the evaluation of the whole hydrolysates bioactivity in a more physiological system (human ex vivo cells) than those usually used (cell-free systems or immortalized cellular lines) with the final purpose of minimizing production costs at the pilot plant level and thus increasing further applicability of this product in the food market.

While the health benefits of wheat gluten protein derivatives have been suggested, to the best of our knowledge, this is the first report of the combined action of WGPHs on both immune and oxidative responses in non-immortalized human cells. Therefore, in addition to the beneficial effects of protein hydrolysation in reducing wheat protein antigenicity [11], our results reinforce the potentially beneficial role of food products containing WGPHs.

## 5. Conclusions

The present study is the first to report that the in vitro administration of Alcalase 2.4 L-generated WGPHs to PBMCs from healthy subjects reduces the production of Th1 and Th17 pro-inflammatory cytokines and increases the anti-inflammatory/pro-inflammatory microenvironment of the PHA-stimulated PBMCs. In addition, WGPHs improve the global antioxidant capacity of the cells by direct free radicals scavenging, increasing the levels of GSH, and counteracting the overproduction of NO. Therefore, these data show that WGPHs might represent an effective option in the functional food generation and support their use in future clinical food trials.

## Figures and Tables

**Figure 1 nutrients-12-01673-f001:**
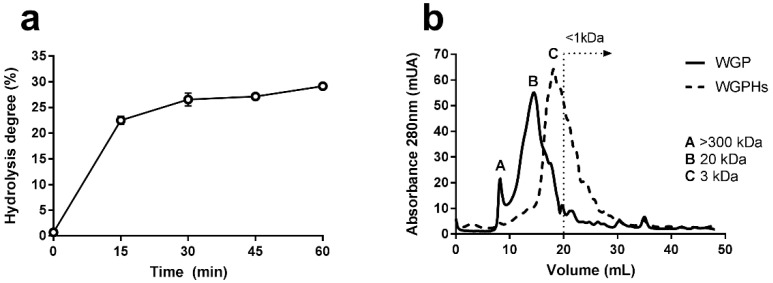
(**a**) Time-course of hydrolysis of WGP by the endopeptidase Alcalase 2.4 L. The data represent the mean and the standard error of the mean of the cleavage of peptide bonds of four determination. (**b**) Molecular weight profiles of WGP and WGPHs obtained by size-exclusion using fast performance liquid chromatography (FPLC). Abs, absorbance at 450 nm; WGP, wheat gluten protein; WGPHs, wheat gluten protein hydrolysates.

**Figure 2 nutrients-12-01673-f002:**
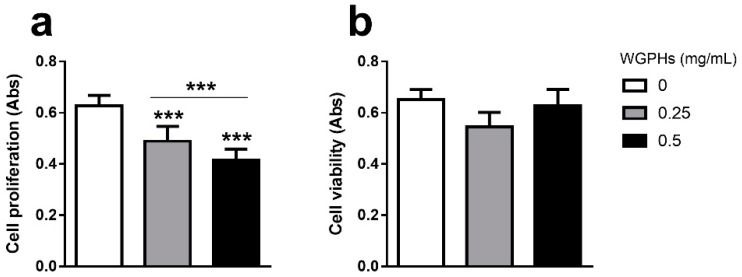
(**a**) Cell proliferation in phytohaemagglutinin-P (PHA)-stimulated cells incubated for 72 h with WGPHs (0, 0.25, and 0.5 mg/mL). (**b**) Cell viability in non-stimulated peripheral blood mononuclear cells (PBMCs) cultured for 72 h with WGPHs (0, 0.25, and 0.5 mg/mL). The data represent the mean and standard error of the mean (*n* = 25). *** *p* ≤ 0.001.

**Figure 3 nutrients-12-01673-f003:**
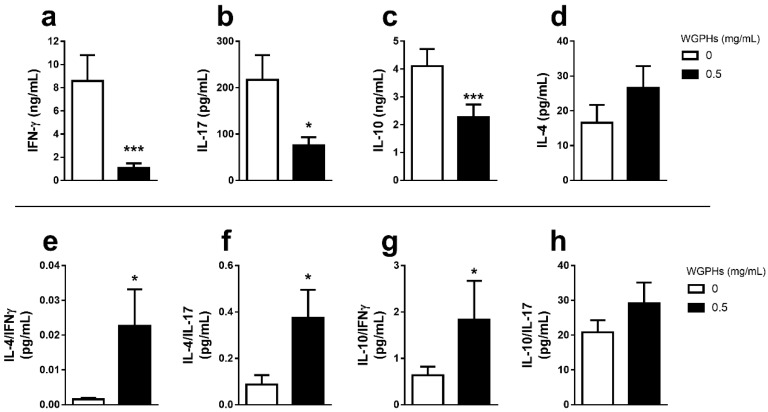
Upper panel: cytokine production in PHA-stimulated PBMCs after 48 h of treatment with/without 0.5 mg/mL WGPHs. Pro-inflammatory IFN-γ (**a**), IL-17 (**b**) and anti-inflammatory IL-10 (**c**), and IL-4 (**d**) cytokine production. Lower panel: ratios of IL-4 (**e** and **f**), IL-10 (**g** and **h**) and pro-inflammatory cytokines (IFN-γ and IL-17). The data represent the mean and standard error of the mean of each group (*n* = 9). * *p* ≤ 0.05; *** *p* ≤ 0.001 with respect to the control group (WGPHs 0 mg/mL).

**Figure 4 nutrients-12-01673-f004:**
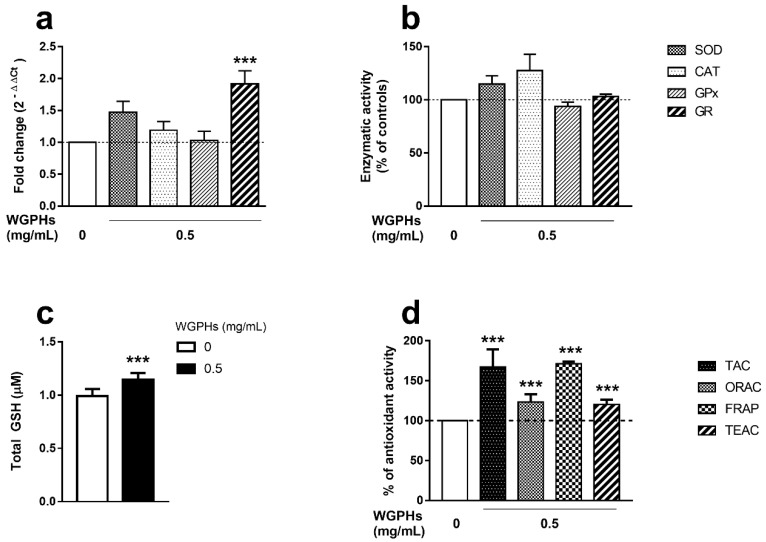
(**a**) Relative gene expression of superoxide dismutase (SOD), catalase (CAT), glutathione peroxidase (GPx), and glutathione reductase (GR) in non-stimulated PBMCs after overnight treatment with 0.5 mg/mL WGPHs. The data represent the mean of the values calculated with respect to PHA-stimulated, untreated cells with the 2^−ΔΔCt^ method, and the standard error of the mean of each group (*n* = 39). (**b**) Enzyme activities of SOD, CAT, GPx, and GR in the supernatants from the same cultures. Activities are expressed as a percentage of the controls. The data represent the mean and standard error of the mean of each group (*n* = 32). (**c**) Total glutathione (GSH) content (µM) in non-stimulated PBMCs after overnight treatment with 0.5 mg/mL WGPHs. The data represent the mean of the GSH content and standard error of the mean of each group (*n* = 28). (**d**) Total antioxidant capacity (TAC), oxygen radical absorbance capacity (ORAC), ferric reducing antioxidant power (FRAP), and trolox equivalent antioxidant capacity (TEAC) quantified on the supernatants from non-stimulated PBMCs cultured overnight with 0.5 mg/mL WGPHs. The data are expressed as a percentage of the control group, and they represent the mean and standard error of the mean of the percentage of each group (*n* = 32). *** *p* ≤ 0.001 with respect to the control group (WGPHs 0 mg/mL).

**Figure 5 nutrients-12-01673-f005:**
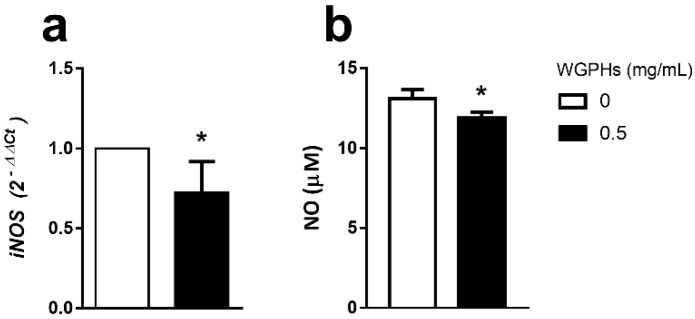
(**a**) Relative gene expression of inducible nitric oxide synthase (iNOS) in PHA-stimulated PBMCs after overnight treatment with 0.5 mg/mL WGPHs. The data represent the mean of the values calculated with respect to PHA-stimulated, untreated cells with the 2^−ΔΔ*C*t^ method, and the standard error of the mean of each group (*n* = 6). (**b**) The concentration of nitric oxide (NO) in the supernatants from overnight non-stimulated cells cultured with 0.5 mg/mL WGPHs. The data represent the mean and standard error of the mean of the percentage of each group (*n* = 32). * *p* ≤ 0.05 with respect to the control group (WGPHs 0 mg/mL).

**Table 1 nutrients-12-01673-t001:** Chemical composition of wheat gluten protein hydrolysates (WGPHs).

%	WGPHs
Protein	73.54 ± 0.44
Moisture	7.97 ± 0.15
Ash	4.60 ± 0.41
Fiber	0.00 ± 0.00
Other ^a^	13.89
Digestibility	100.00 ± 0.00

^a^ Calculated as: 100%—% proteins—% moisture—% ash—% fiber. Data are expressed as mean ± SD of percentage on a dry basis. WGPH, wheat gluten protein hydrolysates.

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
