# Peer review of "Immunomodulatory and Antioxidant Properties of Wheat Gluten Protein Hydrolysates in Human Peripheral Blood Mononuclear Cells"

_nutrients, 2020, doi:10.3390/nu12061673_

Round 1

Reviewer 1 Report

General comments

This study describes inflammatory and antioxidant capacities of an Alcalase-generated wheat gluten protein hydrolysate. Data support potential benefits of the treatment with a modulatory role on Type 1 T helper (Th1) and Th17 pro-inflammatory cytokines. However, it results a little bit masked if the study is intended to contribute to celiac or another disease. My thought is that clarifying this aspect it can help to better understand the scope of the manuscript. The document is written in an adequate language.

Specific Comments

Page 1, line 22-23, line 44: Physiological digestion of gluten proteins results quite inefficient and releases low molecular (tetra)peptides keeping most immunogenic sequences. In my opinion, this sentence may generate misunderstandings. Please, consider rewriting it.

Page 1, line 38: What do authors understand by ‘Inapropiate’?. Imbalances in calorie intake, proportion of macro/micro-nutrients? Low nutritional value of proteins?. Please, could you try to define or consider rewriting it.

Page 1, line 38: What diseases?. Immune- or metabolic-based, loss of organic function?

Page 1, line 42: It has been described the important role of ‘non-gluten’ proteins, such as a-amylase/trypsin inhibitors boosting innate immune response(s) (Junker et al., 2012 J Exp Med 209(13):2395-408). Notably, these compounds are poorly digestible by and exert significant effects at intestinal, but also in the airways.

Page 2, line 57: ‘…inflammatory diseases’. Do authors refer to ‘Celiac Disease’?. What is the antioxidant status in Celiac patients? (Oxid Med Cell Longev. 2018, 2018:1324820) Please, could authors try to be concise.

Page 2, line 60: ‘…health’. Please, be concise. A quick look to the literature pops up sentences such as ‘In some studies, antioxidant supplements did not reduce the risk for cancer or prevent tumour growth; at the contrary, these interventions resulted in some cases to be harmful to the patients’ (Curr Top Med Chem. 2015;15(2):170-8).

Page 2, line 69: ‘…no study…’. J Nutr Biochem. 2018, 54:11-17; Front Nutr . 2019, 6:167.

Page 2, line 76: As stated before, digestion of gluten does not necessarily mean that most immunogenic sequences disappear (Am J Physiol Gastrointest Liver Physiol . 2014, 307(8):G769-76). How can authors assume no response(s) by T cells (most relevant population affecting gluten enteropathy).

Page 3, line 103: Do authors can provide data concerning the immunephenotyping of isolated PBMCs, before and after challenge?. What could be expected in relation to celiac disease? (Front Nutr . 2019, 6:167)

Page 4, line 117: ‘PHA’. Please, spell out completely the first time appeared.

Page 4, line 120: A more precise estimation of cell viability could have been attained by using ‘metabolic’ tests. Here, authors study ‘Cell proliferation (BrdU) and cell viability with a ‘Cell proliferation (WST-1)’ reagent. Perhaps, it is useful to explain (briefly) the advantage of using this ‘double’ evaluation.

Page 5, line 161: PBMCs could include linfomonocytic populations. Monocyte-derived macrophages play important roles as tissue repair effectors and regulators of adaptive immune reactivity. In fact, there is increasing evidence of the continuous flow of classical monocytes (Ly6Chigh) entering the intestine through the CCR2-CCL2 axis, although the contribution of alternative monocytes (Ly6CLow) is less evident. Reducing proliferation, could it be expected negative/positive effects in relation to celiac disease?.

Page 5, line 175: ‘…control…’. Authors have not used positive stimuli for inflammatory processes. Can it be said ‘control’?. Maybe, ‘do not trigger inflamm..’. Please, consider rewriting it.

Page 5, line 179: Why do authors chose IL-17?. IL-15, IL-8 could be more relevant for celiac patients. It could be interesting to reveal the origin of IL-10; Tregs, CD3+, etc.

Page 7, line 211: ‘…supernatants..’. Do authors mean cell extracts?.

Page 7, line 217: ‘…supernatants..’. It is know the interference of low molecular peptides on the measurement of the total antioxidant capacity. How do authors eliminate this effect?.

Page 8, line 242: ‘pepsin and pancreatin’. Physiological digestion implies these enzymes, can these help increasing the gluten digestibility rate?. It can have important consequences for celiac patients.

Page 8, line 249: Do authors refer to ‘Celiac Disease’?. It can be named many different ‘inflammatory diseases’ (i.e., obesity, T2D and cancer) where the inhibition of PBMCs proliferation can be responsible for negative effects.

Page 9, line 290: It could have been very interesting to prove this antioxidant capacity using ‘stressed cells’ to better understand the significance and extent of the results.

Reviewer 2 Report

I am a little uneasy about the sheer number of 11 authors, but I have to largely take the authors' word that each contributed significantly. I can see there were collaborations between several institutions.

Q.Materials and methods line 77, 78:  The investigators say that the gluten isolate was dissolved in water, but gluten by definition is insoluble in water. What do you mean by 'isolate' or was it already partially processed beforehand?

You could  compare with a pepsin chymotrypsin digest, so called Fraser's fraction III. Gluten is mainly digested by pepsin but is relatively resistant to chymotrypsin. But this generates polypeptides that are 'toxic'. Or more recently there is Bob Anderson's work on the 33 mer immunodominant polypeptide and from memory there is a separate polypeptide is responsible for activating intraepithelial lymphocytes to change to a cytotoxic NK phenotype with interleukin 15 production.

Also the latest thing is that the true epitopes may be microbial transglutaminse processed gluten.

Was the blood derived cells HLADQ2 or DQ8? Because these are necessary to activate T-cells that recognise toxic sequences of gliadin. Otherwise no celiac disease.

The manuscripti is nicely composed and set out with good English syntax.
